# Supervised Learning Algorithm Based on Spike Train Inner Product for Deep Spiking Neural Networks

**DOI:** 10.3390/brainsci13020168

**Published:** 2023-01-18

**Authors:** Xianghong Lin, Zhen Zhang, Donghao Zheng

**Affiliations:** College of Computer Science and Engineering, Northwest Normal University, Lanzhou 730070, China

**Keywords:** spiking neural networks, deep learning, spike train inner product, feedback alignment mechanism

## Abstract

By mimicking the hierarchical structure of human brain, deep spiking neural networks (DSNNs) can extract features from a lower level to a higher level gradually, and improve the performance for the processing of spatio-temporal information. Due to the complex hierarchical structure and implicit nonlinear mechanism, the formulation of spike train level supervised learning methods for DSNNs remains an important problem in this research area. Based on the definition of kernel function and spike trains inner product (STIP) as well as the idea of error backpropagation (BP), this paper firstly proposes a deep supervised learning algorithm for DSNNs named BP-STIP. Furthermore, in order to alleviate the intrinsic weight transport problem of the BP mechanism, feedback alignment (FA) and broadcast alignment (BA) mechanisms are utilized to optimize the error feedback mode of BP-STIP, and two deep supervised learning algorithms named FA-STIP and BA-STIP are also proposed. In the experiments, the effectiveness of the proposed three DSNN algorithms is verified on the MNIST digital image benchmark dataset, and the influence of different kernel functions on the learning performance of DSNNs with different network scales is analyzed. Experimental results show that the FA-STIP and BP-STIP algorithms can achieve 94.73% and 95.65% classification accuracy, which apparently possess better learning performance and stability compared with the benchmark algorithm BP-STIP.

## 1. Introduction

The biological brain has a complex structure and function, and carries out information transmission, transformation, and learning under the synaptic plasticity mechanism. Artificial neural networks (ANNs) are abstractions and simulations of the structure and function of biological nervous system, and play an extremely crucial role in signal processing and pattern recognition [1,2]. Spiking neural networks (SNNs) are the new generation of neural computing models, which are composed of more realistic biological spiking neuron models as the basic units [3]. The spike train is the carrier of information representation and processing, and this type of neural encoding method (temporal encoding) integrates multiple aspects of neural information (time, frequency, phase, etc.) [4]. Compared with the traditional ANNs based on frequency encoding, SNNs are more capable of impressive computing power and eminently suitable for complicated spatio-temporal pattern processing. The biological brain with multilayer structures can transform and extract signals layer by layer. According to this idea, researchers have proposed different ANNs with deep structures and their deep learning methods [5,6,7]. Deep learning has led to major advances in image understanding, target detection, speech recognition, natural language processing and other fields, which provides an efficient computational mode for complex pattern recognition problems [8,9]. According to the event-driven network simulation process, the spike-based learning methods of SNNs are considered to be more biologically plausible. It will be a very challenging task in machine learning research to construct the widely applicable supervised learning methods for deep spiking neural networks (DSNNs) combining the spike-based computational mode of SNNs with the signal deep transformation process [10].

The success of traditional deep neural networks is mainly attributed to the application of the well-known error backpropagation (BP) algorithm. However, spiking neurons encode and process information through precisely timed spike trains, and the inherent non-differentiability of DSNNs causes great difficulties in directly using the BP algorithm, which hinders the rapid development and wide application of DSNN learning algorithms. For the design of supervised learning algorithms of DSNNs, some researchers are committed to the exploration and extension of the BP algorithm [11,12]. However, in the BP algorithm, there is a well-known weight transport problem that network connection weights are used to compute both node activations and error gradients for synapses of hidden units [13]. In other words, in the process of error backpropagation, the error signal feedback pathway should be completely consistent with the forward transmission pathway of the network, which is biologically implausible. For that, Lillicrap et al. [14] proposed a feedback alignment (FA) mechanism to alleviate the weight transport problem. Unlike the BP mechanism, FA can transmit teaching signals layer-wise by multiplying the errors with the fixed random synaptic weights. Thus, copies of the synaptic weight matrices in the network are avoided. Moreover, on the basis of FA, Nøkland et al. [15] proposed a simplified error feedback mechanism, called the direct feedback alignment mechanism, also known as broadcast alignment (BA). Using the BA mechanism, the error signals from the output layer are fed back directly to the target layers through the direct-connected random synaptic weight matrices. In this way, even if the network is quite deep, it can propagate the teaching signal nicely, and can effectively avoid the problems of gradient disappearance or gradient explosion that often occur in deep networks [16].

In this study, the spike train will be accurately converted into a continuous function through a given kernel function, which can be interpreted as a specific neurophysiological signal, such as the postsynaptic potential of neurons or the density function of spike emission [17,18]. By using the definition of kernel function and spike train inner product (STIP), the discrete spike signals are mapped into differentiable mathematical space while preserving complete and accurate spiking time information, which satisfies the precondition of derivations of the gradient-based DSNN algorithms. According to different error propagation mechanisms, three algorithms are constructed for DSNNs in this paper: (1) backpropagation based on spike train inner product (BP-STIP), (2) feedback alignment based on spike train inner product (FA-STIP), and (3) broadcast alignment based on spike train inner product (BA-STIP). By constructing a novel error function of spike trains, the BP-STIP learning algorithm is deduced using the mechanism of error backpropagation. The FA-STIP and BA-STIP algorithms are further proposed to realize the avoidance of the weight transport problem through the random feedback pathway, and enhance the stability of learning process as well as improve the learning accuracy of DSNNs.

The rest of this paper is organized as follows. In Section 2, we provide a brief description of the related works, which includes the different supervised learning algorithms for DSNNs. In Section 3, we introduce the definition of the inner product of spike trains, and construct the error function of the network. Section 4 deduces the supervised learning rules based on STIP with the different error signal feedback pathways. Section 5 demonstrates the performance of DSNNs trained with the proposed learning algorithms on the MNIST dataset, and the experimental results are also analyzed and discussed. The conclusion is presented in Section 6.

## 2. Related Works

Spike trains are used to encode and process information in SNNs, and internal state variables of neurons and error functions of the network no longer meet the condition of continuous differentiability. In recent years, more supervised learning algorithms for shallow networks have been proposed [19,20]. However, constructing effective learning algorithms of DSNNs is more important for complex pattern recognition problems, and has attracted more and more attention from researchers. According to the different ideas of learning rules of synaptic weights, the supervised learning algorithms of DSNNs can be divided into three categories [10,21].

### 2.1. Learning Algorithms Based on Network Conversion

The core of ANN-to-SNN conversion approaches is to fully utilize the BP algorithm to accomplish training on conventional deep networks, and then convert the synaptic weights of the fine-trained ANNs to the DSNN computing platforms [22]. Due to the discrepancy in the way of information processing between the two different neural computing platforms, the key point of this type of research is determining how to convert all analog neurons in traditional deep networks into spiking ones in DSNNs. For the conversion from analog neurons without a notion of time to spike-driven ones, the goal is that the DSNN obtains the consistent input-output mapping with the original deep neural network. So far, various ANN-to-SNN conversion approaches have been proposed and achieved good results in the static image classification field.

Diehl et al. [23] analyzed the effects of converting deep ANNs into SNNs, and proposed a conversion method based on weight and threshold balancing. By optimizing the parameters of fully connected DSNNs, the proposed method could minimize performance loss in the conversion process, and was tested on the MNIST dataset. Recently, Kheradpisheh et al. [24] proposed an agent-based DSNN training method and tested it on the Fashion-MNIST and CIFAR-10 datasets. Using the idea of network conversion, researchers converted the training results of traditional convolutional neural networks to convolutional DSNNs, which were more applied to the field of image processing according to their structural characteristics [25,26]. In addition, O’Connor et al. [27] constructed a learning method for spiking deep belief networks (SDBNs) based on the traditional contrastive divergence algorithm. The Siegert approximation for leaky integrate-and-fire (LIF) neurons was used to map an offline-trained deep belief network onto an efficient event-driven SNN. However, the ANN-to-SNN conversion method could not accurately express the temporality of neural information and asynchronous spike transmission, which makes it difficult for DSNNs to learn effectively.

### 2.2. Learning Algorithms Based on Synaptic Plasticity

For SNNs based on temporal encoding, spike trains not only lead to the continuous changes of synapses, but also obey to the spike-timing-dependent plasticity (STDP) mechanism [28]. STDP is a typical unsupervised learning rule, which is also considered to be a crucial mechanism for brain learning and information storage. Using the STDP mechanism to design the supervised learning algorithm of DSNNs is a kind of more biologically plausible learning method. It is an important research direction to apply STDP to the supervised training process of deep network structure.

Lee et al. [29] proposed a convolutional DSNN that is populated with LIF neurons interconnected by shared synaptic weight kernels, and trained convolutional kernels in a layer-wise STDP unsupervised manner. In [30], the convolutional DSNN is trained in two phases. The convolutional kernels are first pre-trained in a layer-wise STDP manner, and the fully-connected layers are then fine-tuned by a spike-based supervised BP method. Bengio et al. [31] introduced a deep learning method using forward and backward neural activity propagation. The learning rules in the DSNN are approximate backpropagation learning based on STDP. By analyzing the approximation between the integrated-and-fire (IF) neuron model and the ReLU activation function, Tavanaei and Maida [32] proposed an approximate BP learning algorithm based on STDP, named BP-STDP. In addition, Liu et al. [33] applied the event-driven continuous STDP learning rule with an adaptive threshold to train DSNNs, and achieved 95~100% classification accuracy in different experimental environments on the video-based face database (MakeFace DB).

### 2.3. Learning Algorithms Based on Gradient Descent

By analyzing the inherent nonlinear mechanism of DSNNs, researchers have developed many deep learning algorithms using the traditional gradient descent method. In the early work, O’Connor and Welling [34] constructed a spiking multilayer perceptron, and designed a BP algorithm using presynaptic and postsynaptic spikes. Lee et al. [35] regarded the membrane potential of spiking neurons as differential signals and discontinuous spike events as noise, and proposed a spike-based BP rule to train DSNN. However, in the calculation of the error function, this method mainly considers the spike frequency fired by neurons. For the feedforward SNNs with a temporal encoding scheme, a supervised learning method based on gradient descent was proposed [36]. The method applies spikes directly to the training of the SNN, which can be used to process spike patterns with complex temporal information. In addition, using the backpropagation through time method of recurrent neural networks, Neftci et al. [37] proposed a surrogate gradient learning algorithm that can effectively train DSNNs.

In the calculation of traditional error backpropagation, the weight adjustment is represented by the feedforward connections of the neural network, while another possible error backpropagation is the FA mechanism. In FA, the feedback pathways are replaced by a group of random connections, that is, the error gradient calculation of each layer depends on the random generated paths. An event-driven random backpropagation (eRBP) algorithm based on a simplified backpropagation pathway was proposed [38]. The learning rule uses error-modulated synaptic plasticity for learning deep representations. Using the BA mechanism, Samadi et al. [39] introduced a supervised learning algorithm in DSNN with fixed feedback weights, in which the learning rules can approximate dynamic input-output relations with piecewise-smooth functions. However, the output layer of DSNN applies one-hot encoding method, and the neurons only have two states of firing or resting, which does not realize the spatio-temporal pattern learning of spike trains. Recently, Shi et al. [40] proposed the DeepTempo method by extending the single-layer Tempotron learning rule to the DSNN. The output errors of each hidden layer are directly computed via the fixed random feedback matrices.

## 3. Definition of STIP and Error Function

Using the specific kernel functions, spike trains can be mapped to the corresponding reproducing kernel Hilbert spaces, in which the inner product operation should be defined. To realize gradient descent calculation, constructing an error function specific to the spike trains is needed, that is, reasonably measure the similarity between the actual output spike trains and the desired spike trains for a DSNN. According to the definition of the error function and the inner product of the spike trains, the learning rules for DSNNs can be deduced.

### 3.1. Kernel Function and STIP Representation

For DSNNs, the spike train $s=\{t^{i}\in \Gamma :i=1,\ldots ,N\}$ is composed of a set of discrete spike times fired by neurons in the simulation interval Γ=[0,T], which can be formally described as [19]:(1)$$s(t)=\sum_{i=1}^{N}\delta (t-t^{i})$$
where $t^{i}$ is the firing time of the *i*-th spike, *N* is the number of spikes in the simulation interval, and δ(·) represents the Dirac delta function. However, when the network error is minimized by gradient descent, it is difficult to define a suitable error between the spike trains using the Dirac delta function. To solve this dilemma, we apply an approach called spike train convolution. Using a specific smoothing function *φ*, each spike train can be uniquely convoluted into a time function [18]:(2)$$s(t)*\varphi (t)=\sum_{i=1}^{N}\varphi (t-t^{i})$$
where *φ* is a finite energy impulse response of a linear filter over time. The resulting transformed spike train is a function in Hilbert space.

To facilitate the subsequent algorithm derivation, the vector $h_{p}(t)$ is defined to represent the mapping of the spike trains generated by the neurons in layer *p* on the continuous time functions:(3)$$h_{p}(t)=\left(\begin{array}{c} s_{p}^{1}(t)*\varphi (t)\\ s_{p}^{2}(t)*\varphi (t)\\ \vdots \\ s_{p}^{H_{p}}(t)*\varphi (t) \end{array}\right)=\left(\begin{array}{c} \sum_{i=1}^{N_{p}^{1}}\varphi (t-t^{i})\\ \sum_{i=1}^{N_{p}^{2}}\varphi (t-t^{i})\\ \vdots \\ \sum_{i=1}^{N_{p}^{H_{p}}}\varphi (t-t^{i}) \end{array}\right)$$
where $s_{p}^{i}(t)$ represents the spike train fired by the *i*-th neuron in layer *p*, $N_{p}^{i}$ represents the total number of spikes of $s_{p}^{i}(t)$ during the simulation interval, and $H_{p}$ is the number of neurons in layer *p* of the DSNN.

The STIP is composed of the inner products of two spike times, which needs to be defined first. Using positive-definite, symmetric and shift-invariant kernel function *κ* (commonly chosen Gaussian function, Laplacian function, etc.), the inner product operation between spike times $t^{m}$ and $t^{n}$ is defined as:(4)$$\kappa (t^{m},t^{n})=\int_{\Gamma }\varphi (t-t^{m})\varphi (t-t^{n})dt$$

Thus, the inner product operation between the spike trains fired by the *i*-th neuron in layer *p* and the *j*-th neuron in layer *q* of the network can be defined as [16]:(5)$$F(s_{p}^{i},s_{q}^{j})=\sum_{m=1}^{N_{p}^{i}}\sum_{n=1}^{N_{q}^{j}}\int_{\Gamma }\varphi (t-t^{m})\varphi (t-t^{n})dt=\sum_{m=1}^{N_{p}^{i}}\sum_{n=1}^{N_{q}^{j}}\kappa (t^{m},t^{n})$$

In other words, the STIP between spike trains $s_{p}^{i}$ and $s_{q}^{j}$ is the accumulation form of spike time pairs with order $O(N_{p}^{i}N_{q}^{j})$. In the simulation interval Γ, the inner product matrix between the spike train vector $s_{p}$ generated by the neurons in layer *p* and the spike train vector $s_{q}$ generated by the neurons in layer *q* can be represented as:(6)$$F(s_{p},s_{q})=\int_{\Gamma }h_{p}(t)h_{q}{\left(t\right)}^{T}dt=\left[\begin{array}{ccc} F(s_{p}^{1},s_{q}^{1}) & \cdots & F(s_{p}^{1},s_{q}^{H_{q}})\\ \vdots & & \vdots \\ F(s_{p}^{H_{p}},s_{q}^{1}) & \cdots & F(s_{p}^{H_{p}},s_{q}^{H_{q}}) \end{array}\right]$$
where $h_{q}{\left(t\right)}^{T}$ represents the transpose of a vector, and $F(s_{p},s_{q})$ is a $H_{p}\times H_{q}$ dimension matrix. Each element in the matrix in Equation (6) represents the inner product operation result of the two corresponding spike trains, as shown in Equation (5).

### 3.2. Spike Train Relationship and Error Function

Spike trains are used to encode neural information and external stimulus signals, so the input and output of spiking neuron are expressed in the form of discrete spike trains. The spiking activity of the postsynaptic neuron is defined by accumulating the spike trains of the presynaptic neurons. The relationship between spike trains of neurons in adjacent layers of the DSNN can be simplified to a linear combination [41]:(7)$$h_{l}(t)=W_{l}h_{l-1}(t)$$
where $W_{l}\in {ℜ}^{H_{l}\times H_{l-1}}$ is the synaptic weight matrix between two adjacent layers in the DSNN. As shown in Equation (3), $h_{l}(t)$ and $h_{l-1}(t)$ represent the continuous functions corresponding to the spike trains generated by the neurons in adjacent layers, respectively. In fact, after the discrete spike train is converted into a continuous function by a specific kernel function, the continuous function of spike strain fired by postsynaptic neuron can be expressed as the linear combination of the continuous functions of spike trains inputted by presynaptic neurons [42].

A crucial step of constructing supervised learning algorithms of DSNNs is to make the reasonable definition of the error function of spike trains, and the least square error is used to measure the similarity of spike trains in this paper. The instantaneous error at time *t* for the network can be expressed as:(8)$$E(t)=\frac{1}{2}{\left\|h_{o}^{a}(t)-h_{o}^{d}(t)\right\|}^{2}$$
where $\left\|h_{o}^{a}(t)-h_{o}^{d}(t)\right\|$ is the vector module, $h_{o}^{a}(t)$ is the continuous functions corresponding to the actual spike trains $s_{o}^{a}(t)$ generated by the output neurons, and $h_{o}^{d}(t)$ represents the continuous functions corresponding to the target spike trains $s_{o}^{d}(t)$. The overall error of the network is the integral of E(t) in the simulation interval Γ:(9)$$E=\int_{\Gamma }E(t)dt$$

Using this error function, the synaptic weight adjustment rules of an arbitrary layer can be derived, so that the error BP mechanism can be utilized under the superiority of preserving the spatio-temporal information of all spike trains.

## 4. DSNN Supervised Learning Algorithms

In this paper, the DSNN adopts a feedforward architecture, where the neurons between the adjacent layers are fully connected. The network includes an input layer, multiple hidden layers, and an output layer. The layers of the network are numbered from the input layer forward, that is, the input layer is numbered as layer 0, the hidden layers are sequentially numbered from layer 1 to *L* − 1, and the output layer is numbered as layer *L*, as shown in Figure 1. Consequently, $W_{l}\in {ℜ}^{H_{l}\times H_{l-1}}$ is the synaptic weight matrix in adjacent layers, and each neuron in layer *l* is postsynaptic to all presynaptic neurons in layer *l* − 1.

In the BP mechanism, the error signal feedback pathway must be strictly equal to the connection weights in the corresponding forward pathway. That is, the dual use of the same connection weights constitutes a bidirectional flow of information through synapses in the network. From a biological point of view, the realization of such a precise neural circuit in the brain is implausible and extremely difficult. Concerning this issue, the effective FA mechanism replaces the feedforward weight matrices in the error signal backpropagation with fixed random weight matrices [14,15]. Figure 1 illustrates the error signal backpropagation mode of FA and BA mechanisms, and the error signal is the result of similarity measurement between the actual spike trains fired by the output neurons and the desired spike trains encoded from the labels. In FA error backpropagation, the forward connection weights are replaced by random synaptic weights, and the error gradient of each layer is computed by accumulative multiplication, as shown in the yellow feedback pathway in Figure 1. The BA is a simpler feedback mechanism, the teaching signal is directly fed back to the target layer through a fixed random feedback matrix (red feedback pathway in Figure 1).

### 4.1. Backpropagation Learning Rule Based on STIP

Similar to the derivation of the BP algorithm of traditional ANNs, our learning algorithm requires to calculate the gradient of the error function to each weight to obtain the synaptic weight adjustment increment. Using the generalized delta update rule, the synaptic weight adjustment rule is expressed as follows:(10)ΔW=−η∇E
where *η* represents the learning rate of the neural network. For weight updating of each layer in the DSNN, the gradient matrix of synaptic weights could be represented as the integration of derivative of the instantaneous error function E(t) with respect to the weight matrix W in the simulation interval Γ, which can be written as:(11)$$\nabla E=\int_{\Gamma }\frac{\partial E(t)}{\partial W}=\int_{\Gamma }\frac{\partial E(t)}{\partial h(t)}\frac{\partial h(t)}{\partial W}dt$$

In order to simplify the derivation of learning rules, the partial derivative ∂E(t)/∂h(t) is marked as δ, which represents the error signal transmitted layer by layer.

Because DSNN is updated by the BP mechanism, the weights are updated from back to front, so the weights of the output layer are updated first. For the gradient calculation of the weight matrix $W_{L}$ between neurons in the output layer and the last hidden layer, the derivative of E(t) to $W_{L}$ can be derived by the chain rule:(12)$$\frac{\partial E(t)}{\partial W_{L}}=\frac{\partial E(t)}{\partial h_{o}^{a}(t)}\frac{\partial h_{o}^{a}(t)}{\partial W_{L}}$$

According to Equation (8), the error signal in layer *L* of the DSNN can be computed as:(13)$$\delta_{L}=\frac{\partial E(t)}{\partial h_{o}^{a}(t)}=\frac{\partial \left(\frac{1}{2}{\left\|h_{o}^{a}(t)-h_{o}^{d}(t)\right\|}^{2}\right)}{\partial h_{o}^{a}(t)}=h_{o}^{a}(t)-h_{o}^{d}(t)$$

Using the linear combination relationship of spike trains in Equation (7), the second partial derivative term of the right-hand part of Equation (12) is computed as:(14)$$\frac{\partial h_{o}^{a}(t)}{\partial W_{L}}=\frac{\partial W_{L}h_{L-1}(t)}{\partial W_{L}}=h_{L-1}{\left(t\right)}^{T}$$

By combining Equations (13) and (14), the derivative of the error function E(t) at time *t* can be obtained. Using Equation (6), the error gradient $\nabla E_{L}$ for adjusting the weight matrix $W_{L}$ in the simulation interval Γ can be calculated as:(15)$$\nabla E_{L}=\int_{\Gamma }\frac{\partial E(t)}{\partial W_{L}}dt=\int_{\Gamma }\left[h_{o}^{a}(t)-h_{o}^{d}(t)\right]h_{L-1}{\left(t\right)}^{T}dt=F(s_{o}^{a},s_{L-1})-F(s_{o}^{d},s_{L-1})$$
where $s_{L-1}(t)$ represents the spike strains fired by the last hidden layer neurons. The weight adjustment rule between the output layer and the last hidden layer can be expressed as:(16)$$\Delta W_{L}=-\eta \nabla E_{L}=-\eta \left[F(s_{o}^{a},s_{L-1})-F(s_{o}^{d},s_{L-1})\right]$$

As seen in Equation (16), the synaptic weight adjustments can be represented in the form of STIP.

Similarly, the error gradient for adjusting the synaptic weights in hidden layer *l* (1≤l≤L−1) can be deduced by the chain rule. The derivative of the instantaneous error E(t) to the weight matrix $W_{l}$ between the postsynaptic layer *l* and the presynaptic layer *l* − 1 is computed as:(17)$$\frac{\partial E(t)}{\partial W_{l}}=\frac{\partial E(t)}{\partial h_{l}(t)}\frac{\partial h_{l}(t)}{\partial W_{l}}$$

The first partial derivative term of the right-hand part of Equation (17) is the error signal in the layer *l*, it can be computed using the error signal $\delta_{l+1}=\partial E(t)/\partial h_{l+1}(t)$ in layer *l* + 1:(18)$$\delta_{l}=\frac{\partial E(t)}{\partial h_{l}(t)}=W_{l+1}{}^{T}\delta_{l+1}$$
where $W_{l+1}{}^{T}=\partial h_{l+1}(t)/h_{l}(t)$ is obtained using Equation (7). The second partial derivative term of the right-hand part of Equation (17) can be computed as:(19)$$\frac{\partial h_{l}(t)}{\partial W_{l}}=\frac{\partial \left(W_{l}h_{l-1}(t)\right)}{\partial W_{l}}=h_{l-1}{\left(t\right)}^{T}$$

By combining Equations (18) and (19), the instantaneous partial derivative for $W_{l}$ can be calculated as:(20)$$\frac{\partial E(t)}{\partial W_{l}}=W_{l+1}{}^{T}\delta_{l+1}h_{l-1}{\left(t\right)}^{T}$$

It can be iteratively solved for Equation (20) using error signals from layer *l* + 1 to layer *L*, Equation (20) is rewritten as:(21)$$\begin{array}{cc} \frac{\partial E(t)}{\partial W_{l}} & =W_{l+1}{}^{T}W_{l+2}{}^{T}\cdots W_{L}{}^{T}\left[h_{o}^{a}(t)-h_{o}^{d}(t)\right]h_{l-1}{\left(t\right)}^{T}\\ & ={\left(W_{L}\cdots W_{l+2}W_{l+1}\right)}^{T}\left[h_{o}^{a}(t)-h_{o}^{d}(t)\right]h_{l-1}{\left(t\right)}^{T} \end{array}$$

The total weight adjustment can be calculated by the integration of the above equation in the simulation interval, the error gradient for adjusting the synaptic weights in layer *l* is computed as:(22)$$\begin{array}{cc} \nabla E_{l} & =\int_{\Gamma }\frac{\partial E\left(t\right)}{\partial W_{l}}dt={\left(W_{L}\cdots W_{l+2}W_{l+1}\right)}^{T}\int_{\Gamma }\left[h_{o}^{a}(t)-h_{o}^{d}(t)\right]h_{l-1}{\left(t\right)}^{T}dt\\ & ={\left(W_{L}\cdots W_{l+2}W_{l+1}\right)}^{T}\left[F(s_{o}^{a},s_{l-1})-F(s_{o}^{d},s_{l-1})\right] \end{array}$$
where $s_{l-1}(t)$ is the spike trains fired by the neurons in layer *l* − 1. Therefore, the synaptic weight update rule between layer *l* and layer *l* − 1 is:(23)$$\Delta W_{l}=-\eta {\left(W_{L}\cdots W_{l+2}W_{l+1}\right)}^{T}\left[F(s_{o}^{a},s_{l-1})-F(s_{o}^{d},s_{l-1})\right]$$

The BP-STIP learning rules are given in Equations (16) and (23) for updating the synaptic weights in the different layers of DSNN, which make full use of the spike trains fired by neurons in the network. The STIP in Equations (16) and (23) can be represented as the accumulation form of the inner product of spike time pairs using Equation (5). However, the synaptic update rules of the BP-STIP algorithm depend heavily on the weight matrices that are used to forward propagate information in the network, which can be observed from Equation (23).

### 4.2. Feedback Alignment Learning Rule Based on STIP

In the BP-STIP algorithm, the error signal is fed back to the target layer (layer *l*) through the transposition of the feedforward weight matrix W, so as to adjust the connection weights between neurons in the DSNN. The layer-wise feedback pathway of the BP-STIP learning algorithm could be represented as: (24)$$path\;(l)={\left(W_{L}\cdots W_{l+2}W_{l+1}\right)}^{T}$$

The error signal and pathway information jointly determine the teaching signal δ transported to each layer. The teaching signal of the BP mechanism can be expressed as:(25)$$\delta_{\mathrm{BP}}(l)={\left(W_{L}\cdots W_{l+2}W_{l+1}\right)}^{T}E$$
where $E=h_{o}^{a}(t)-h_{o}^{d}(t)$ is an error vector for all output neurons in the network.

FA mechanism indicates that accurate feedback pathway is not a necessary condition in the learning process. The yellow dotted arrows in Figure 1 illustrate the propagation pathway of error signal of the FA mechanism. In this way, the error signal can be propagated through the randomly layer-wise pathway, and the teaching signal transported to the layer *l* in the DSNN can be expressed as:(26)$$\delta_{\mathrm{FA}}(l)={\left(B_{L}\cdots B_{l+2}B_{l+1}\right)}^{T}E$$
where $B_{l}\in {ℜ}^{H_{l}\times H_{l-1}}$ represents the feedback matrix randomly generated between adjacent layers, and its dimension is consistent with $W_{l}$. The error signal feedback pathway in the FA mechanism is separated from the forward pathway.

According to the error backpropagation pathway given in Equation (26), the FA-STIP learning algorithm of DSNNs can be constructed. The adjustment of synaptic weights between the output layer and the last hidden layer is calculated according to Equation (16), and the adjustment rule of other layers can be modified as:(27)$$\Delta W_{l}=-\eta {\left(B_{L}\cdots B_{l+2}B_{l+1}\right)}^{T}\left[F(s_{o}^{a},s_{l-1})-F(s_{o}^{d},s_{l-1})\right]$$

In the FA-STIP algorithm, the adjustment of synaptic weights of each layer is no longer affected by the forward pathway of the DSNN, but determined by the error signal calculated at the output layer and the randomly generated feedback matrix of each layer, which can effectively improve the stability and flexibility of network training.

### 4.3. Broadcast Alignment Learning Rule Based on STIP

Compared with the BP-STIP learning algorithm, the FA-STIP error feedback mechanism no longer strictly depends on the forward pathway of the network, but the error signal still needs to be fed back layer-wise, which still lacks a reasonable explanation in biology. The researchers further simplified the FA mechanism and proposed a BA mechanism that eliminates the limitation of layer-wise error signal transmission [15]. As shown by the red dotted arrows in Figure 1, the error signal is broadcast directly to the neurons of the target layer. The error signal feedback mode of the BA mechanism is only determined by the information of the target layer and has the local learning characteristics of synaptic weight adjustment. The teaching signal of the BA mechanism can be represented as:(28)$$\delta_{\mathrm{BA}}(l)={\left(B_{l}^{\prime }\right)}^{T}E$$
where $B_{l}^{\prime }\in {ℜ}^{H_{L}\times H_{l}}$ represents the randomly generated feedback matrix between the output layer and the target layer, and $H_{L}$ is the number of neurons in the output layer, and $H_{l}$ is the number of neurons in layer *l*.

FA-STIP is further extended by using the BA feedback mechanism, so that the error signal can remove the bondage of layer-wise propagation. Similarly, the adjustment of the synaptic weights between the output layer and the last hidden layer is calculated based on Equation (16). According to the error broadcast pathway in Equation (28), the synaptic weight adjustment rules of other layers can be expressed as:(29)$$\Delta W_{l}=-\eta {\left(B_{l}^{\prime }\right)}^{T}\left[F(s_{o}^{a},s_{l-1})-F(s_{o}^{d},s_{l-1})\right]$$

In the BA-STIP algorithm, the adjustment of synaptic weights in each layer only depends on the local spiking activity information of the output layer and presynaptic neurons, as well as the global feedback signal. The BA learning rule is also known as the three-factor synaptic plasticity rule [43]. In this way, even if the DSNN structure is very deep, the error signal can be propagated quickly and effectively, and it is more in line with the local signal driven learning mechanism of the biological brain.

FA-STIP and BA-STIP learning algorithms based on the FA mechanism and its extension offer a good solution for the weight transport problem, and are more biologically plausible than the error backpropagation process of the BP-STIP algorithm. According to experimental results, it is found that the learning algorithms based on FA and BA mechanisms have better learning stability and can effectively avoid the problem of gradient explosion and disappearance during the DSNN training process.

## 5. Experimental Results and Discussion

### 5.1. Deep Learning Framework and Parameter Settings

In order to verify the learning performance of BP-STIP, FA-STIP and BA-STIP deep learning algorithms in a practical pattern classification problem, this paper selects the MNIST handwritten digital image dataset, which is a benchmark dataset that widely used in the field of machine learning. The MNIST dataset includes 70,000 digital handwritten grey-scale images with 28 × 28 pixels, in which the image is labelled from 0 to 9. In the experiment, 60,000 items are randomly picked out for training and the remaining for testing.

Figure 2 illustrates the process of MNIST classification based on the DSNN with temporal encoding. Unlike traditional deep neural networks, spike trains are used to encode and process information. This classification framework can be divided into five submodules:Spike train encoding of grey-scale images. The first step is to flatten the two-dimensional matrix of image data into a one-dimensional vector with 784 pixels, and normalize elements of the vector to [0,1]. Then, 784 spike trains are generated from the pixel vector of the sample image using the Poisson encoding method [44]. Subsequently, these spike trains are inputted into the input layer of DSNN.DSNN simulation. The clock-driven strategy is applied to simulate the DSNN with multiple hidden layers, and the spike trains fired by neurons are used for synaptic weight learning and sample classification.Decoding and sample classification. According to the number of categories of MNIST problems, the output layer of the network contains 10 neurons, and the spike trains of output neurons are used for decoding and classification. Similar to the calculation method of Softmax, the category is divided according to the similarity between the spike trains of output neurons and the target spike trains.Target spike train generation based on labels. The DSNN takes spike trains as the carrier of information processing and propagation, so labels from 0 to 9 need to be encoded into corresponding target spike trains, which are important information for sample classification and error signal calculation.Error backpropagation and deep learning. According to the actual spike trains of the output layer neurons, combined with the target spike trains, the error of DSNN is calculated. Based on the different feedback pathways of error signal backpropagation, the proposed deep learning algorithm, such as BP-STIP, FA-STIP or BA-STIP, is applied to adjust the synaptic weights of the network.

In the experiment, each label corresponds to one target neuron in the output layer of the DSNN. Under the ideal condition, when the input sample corresponding to the assigned label is inputted in the network, the target neuron is activated and emits spikes evenly, while other neurons in the output layer remain rest during the simulation interval. Therefore, the target spike trains are generated through the linear encoding method [45], and the detailed label encoding process is shown in Figure 3. Label data are first processed by the one-hot encoder, and the encoded data are then laterally extended to target spike trains of 10 neurons in the output layer over the simulation interval.

In this paper, a four-layer feedforward network is used in the image classification, which includes an input layer composed of 784 neurons, two hidden layers, and an output layer composed of 10 neurons. A LIF neuron model is chosen as the basic computing unit, the parameters of all LIF neurons in the network are set as follows: the time decay constant of postsynaptic potential is 5.0 ms, the threshold of spike firing is 5.0, and the resting and reset potential is set to 0. Additionally, the simulation interval of the network is set to 10 ms, and the initial matrices W of synaptic weights as well as the feedback matrices B and $B^{\prime }$ in the backpropagation process are randomly generated, and satisfy the normal distribution of N(0,1). In the learning stage, the DSNN is trained with 150 iterations for each experiment, the batch size is fixed to 64, and the learning rate is set to 0.00008. Each experimental result was recorded as the average of 10 trials. The parameter settings in the experiment are summarized in Table 1. The MNIST digital image classification system was implemented in a Python environment using the open-source deep learning SNN framework Spikingjelly based on PyTorch. All experiments were carried out on an Intel(R) Xeon(R) Gold 5218 CPU and a NVIDIA Quadro RTX 6000 GPU.

### 5.2. Learning Process Analysis of the Algorithms

In order to compare and analyze the learning performance of BP-STIP, FA-STIP, and BA-STIP learning algorithms in DSNNs, the first group of experiments adopts the network structure with 784-800-800-10. The STIP in synaptic weight learning rules is calculated by the Gaussian kernel function, and the network is adjusted according to the updated weight values. On the MNIST dataset, after 150 iterations of learning of an experiment, the trend of classification accuracy of DSNN on the training set and testing set is shown in Figure 4.

Figure 4a shows the learning performance of the BP-STIP algorithm. After four iterations, the classification accuracy of DSNN reaches 90.82% on the training set and 91.48% on the testing set. When the number of iterations is between 5 and 75, the classification accuracy of the algorithm fluctuates greatly during training. From the 75th iteration to the 150th iteration, we note that the classification accuracy increases slowly, and then tends to be gradually stable. The BP-STIP learning algorithm depends heavily on the synaptic weight information of the forward pathway, thus the algorithm is characterized by high instability during the learning process. Accordingly, the trained DSNN has extremely weak classification performance. Applying the FA-STIP algorithm with the same network parameter settings, the learning performance trend of the network is shown in Figure 4b. The classification accuracy increases rapidly when the number of iterations is less than 25, and after that, although the training accuracy of DSNN is still increasing slowly, the testing accuracy is no longer significantly changed. After 150 iterations, the classification accuracy reaches 94.13% for the training set and 93.02% for the testing set. Compared with BP-STIP, the FA-STIP learning algorithm possesses high stability in the learning process due to the replacement of the weight matrices W by the randomly generated feedback matrices B in the synaptic weight adjustment rules. Under the same conditions, the classification accuracy of the BA-STIP algorithm can be obtained and shown in Figure 4c. After 150 iterations of DSNN, the accuracy of the training set is 91.85%, and the highest accuracy of the testing set reaches 92.37%.

As can be seen from the learning process in Figure 4, FA-STIP and BA-STIP, two learning algorithms based on the feedback alignment mechanism, have better convergence ability for the MNIST dataset than the BP-STIP algorithm based on the traditional BP mechanism. Using the randomly generated feedback matrices to update synaptic weights, FA-STIP and BA-STIP algorithms can be independent of the feedforward propagation pathway of DSNNs. Therefore, the two algorithms reduce the violent fluctuation of classification accuracy during the learning process, and effectively avoid the gradient explosion and disappearance problem. Moreover, the BA-STIP algorithm propagates the error signal through a direct feedback pathway, which accelerates the convergence speed of the network to a certain extent.

### 5.3. Comparison and Analysis of Different Kernel Functions

The second group of experiments is used to test the classification effect of three learning algorithms on the MNIST dataset under the four-layer DSNN structure with different kernel functions and different scales. Using different kernel functions, discrete spike trains can be mapped to different reproducing kernel Hilbert spaces [17]. To verify the influence of different kernel functions on the learning performance of our algorithms, the Gaussian, Laplacian, Inverse multiquadratic, and α-kernel functions are chosen for the STIP calculation in the learning rules. The parameters of different kernel functions are adjusted through the learning performance of the algorithms, and the resulting kernel parameter settings are listed in Table 2. In addition, the number of neurons in the hidden layers increases from 400 to 1000 with an interval of 200, while other parameter settings of the network remain the same. 

The BP-STIP, FA-STIP, and BA-STIP algorithms using the Gaussian kernel function are trained through 150 iterations, and the trend of classification accuracy of DSNN with increasing number of hidden layer neurons is shown in Figure 5. For the Gaussian kernel-based BP-STIP algorithm, the classification accuracy increases first and then decreases with the increase of the number of hidden layer neurons. When the network structure is set to 784-800-800-10, the classification accuracy reaches 90.82% for the training set and 91.48% for the testing set. For the FA-STIP algorithm with Gaussian kernel, the classification accuracy shows an obvious upward trend as the number of hidden layer neurons increases. When the DSNN scale is 784-1000-1000-10, the classification accuracy reaches the highest, with 94.02% on the training set and 94.73% on the testing set. Moreover, the Gaussian kernel-based BA-STIP algorithm achieves its optimum classification result when the number of hidden layer neurons is 400, with a training accuracy of 91.62%, and a testing accuracy of 92.71%, and then exhibits a declining trend in accuracy as the number of hidden layer neurons increases. Using the Gaussian kernel to calculate STIP, the learning effect of the FA-STIP algorithm is better than BP-STIP and BP-STIP algorithms on the MNIST classification task. Despite the classification accuracy of the BA-STIP learning algorithm is inferior to the FA-STIP algorithm, its iteration number to reach the convergence state is less than that of the FA-STIP algorithm, and the time required for one iteration training is also shorter than that of the FA-STIP algorithm. Therefore, the Gaussian kernel-based BA-STIP learning algorithm can realize rapid learning of image classification problems.

The average classification accuracy obtained after 150 iterative training of BP-STIP, FA-STIP and BA-STIP algorithms with the Laplacian kernel function is shown in Figure 6. When the network scale increases, the BP-STIP algorithm based on the Laplacian kernel has no obvious change trend in the classification accuracy of DSNN on the MNIST dataset. The structure of DSNNs is 784-800-800-10, and the best classification accuracy on the training set and the testing set reaches 90.13% and 91.56%, respectively. The learning capacity of the FA-STIP algorithm increases with the expansion of the network scale. When the number of hidden layer neurons is set to 1000, the learning performance reaches the highest, with an accuracy of 94.33% on the training set and 94.75% on the testing set. Conversely, the classification accuracy of the BA-STIP learning algorithm decreases as the number of hidden layer neurons gradually increases. When the DSNN scale is set to 784-400-400-10, the BA-STIP algorithm has better learning performance, the accuracy of the training set is 94.52%, and the accuracy of the testing set is 94.69%. Therefore, the BA-STIP learning algorithm based on the Laplacian kernel function is more suitable for small-scale neural networks.

Through the experiments, it is found that the BP-STIP learning algorithm constructed by using the inverse multiquadratic kernel function is incapable of processing and analyzing information, and cannot complete feature learning on MNIST dataset, but the FA-STIP and BA-STIP learning algorithms based on this kernel function exhibit learning ability. After 150 iterations of FA-STIP or BA-STIP algorithms based on inverse multiquadratic kernel function, the network classification performance is shown in Figure 7. With the increasing of the number of hidden layer neurons in the DSNN, the learning effect of both learning algorithms shows an upward trend. In addition, the classification accuracy of the FA-STIP algorithm is better than that of the BA-STIP algorithm when the network scale is relatively large. When the number of hidden layer neurons is set to 400 and 600, the learning performance of the BA-STIP algorithm is better than the FA-STIP algorithm. When the number of hidden layer neurons is 800 and 1000, the learning accuracy of the FA-STIP algorithm is higher. In general, the FA-STIP and BA-STIP algorithms based on inverse multiquadratic kernel have learning ability on complex datasets, but their learning performance is poor compared with those algorithms based on Gaussian kernel and Laplacian kernel functions.

Similarly, using the BP-STIP algorithm based on the α-kernel function to classify the MNIST dataset, the experiment found that it is unable to complete the effective learning and classification of this task. While the corresponding FA-STIP and BA-STIP algorithms can learn the features of the MNIST dataset and complete classification, and the accuracy under different network scales is shown in Figure 8. Based on the α-kernel function, the learning performance of the BA-STIP algorithm is significantly better than the FA-STIP algorithm. However, with the increment of the number of hidden layer neurons in DSNNs, the classification accuracy of the FA-STIP algorithm still maintains the trend of growth, and that of the BA-STIP algorithm shows a constantly decreasing trend. The FA-STIP algorithm achieves the highest classification accuracy under the network structure of 784-1000-1000-10, with 94.07% on the training set and 94.16% on the testing set. When the network scale is 784-400-400-10, the classification accuracy of the BA-STIP algorithm is 95.80% for the training set, and 95.65% for the testing set. The BA-STIP algorithm based on the α-kernel function achieves good learning ability under the current network settings.

Through the pattern classification experiment on the MNIST dataset, the processing ability of the proposed three STIP-based deep learning algorithms for practical complex problems is verified, and the influence of different kernel functions and different network scales on learning performance is analyzed. The experimental results show that the BP-STIP algorithm, which depends on the forward pathway of the network, has poor stability during the learning process, and even fails to realize the feature learning and classification on the MNIST dataset under the inverse multiquadratic kernel and α-kernel functions. While the FA-STIP and BA-STIP algorithms propagate error signals through randomly generated feedback matrices, which have good learning stability, as well as solving ability for complex problems. Overall, the FA-STIP algorithm shows better learning ability in large-scale networks with more hidden layer neurons, while the BA-STIP algorithm shows more prominent learning ability in small-scale networks. Furthermore, the BA-STIP learning rule is simpler than the BP-STIP and FA-STIP algorithms, and the calculation time consumed in the weight adjustment of DSNN is less.

### 5.4. Comparison of Different Deep Learning Algorithms

Based on STIP operation, this paper proposes three DSNN learning algorithms with different error feedback mechanisms, the experimental classification accuracies obtained are compared against several other deep learning algorithms. Table 3 shows the comparison results of classification accuracies of different methods on the MNIST dataset. When the network scale is 784-800-800-10, the BP-STIP learning algorithm with the Laplacian kernel function achieves the best classification accuracy of 91.56% on the testing set. For the two DSNN learning algorithms based on the feedback alignment mechanism, when the network scale is 784-1000-1000-10 and the Gaussian kernel function is adopted, the classification accuracy of the FA-STIP algorithm on the MNIST dataset is 94.73%; when the network scale is 784-400-400-10 and the α-kernel function is used, the classification accuracy of BA-STIP algorithm on MNIST dataset achieves 95.65%.

The DSNNs trained by ANN-to-SNN conversion methods can achieve relatively higher classification accuracy on the MNIST dataset [23,25,26]. However, more networks use deep convolutional structures based on the IF neuron model, and *x*C represents the convolutional layer containing *x* channels, and P represents the pooling layer in Table 3. This kind of learning algorithm uses frequency encoding, which cannot accurately express the temporality and asynchronous transmission of spikes, nor can it directly train DSNNs. In addition, the learning algorithm of the SDBN based on the spike event-driven mode achieves 94.09% classification accuracy on the MNIST dataset [27]. Using the STDP mechanism to train DSNNs, the unsupervised layer-wise STDP learning method achieved a classification accuracy of 91.1% on the MNIST dataset for convolutional DSNNs [29], and further work has considered the supervised BP fine-tuning process [30]. When the network structure is 784-500-150-10, the classification accuracy of the BP-STDP algorithm with spike frequency encoding is 97.2% [32]. More works have employed the end-to-end BP method to train DSNNs directly [35,36]. Although these BP-based methods achieved relatively high classification accuracy, the derivation of learning algorithms depends on the change process of membrane potential of specific spiking neurons, such as the LIF neuron model, and could not be adapted to networks with arbitrary spiking neuron models. In [38,39,40], the BA mechanism was applied to construct the supervised learning algorithm of DSNNs, the training time of the networks can be effectively reduced through the direct feedback of error signal. On the premise that the temporal encoding is used in DSNNs, the FA-STIP and BA-STIP learning algorithms proposed in this paper are comparable with the DeepTempo method for classification accuracy on the MNIST dataset. When the network structure of DSNN is 784-400-400-10, the classification accuracy of BA-STIP is 95.65%, and that of DeepTempo is 95.1%.

Furthermore, it can be summarized that: (1) the learning methods based on frequency encoding achieve high classification accuracy on the MNIST dataset, mainly including ANN-to-SNN conversion methods, and BP methods by approximating or restricting the spiking neuron models. (2) The classification accuracy of convolutional DSNNs is higher than that of feedforward connected networks, because the network structure with convolutional and pooling layers is more suitable for learning from image datasets. (3) Although the learning methods based on temporal encoding are more biologically plausible, such methods have lower classification accuracy than the learning methods based on spike frequency encoding. One of the reasons is the lack of effective spike encoding methods for the data of practical problems. The learning algorithms we built also suffer from the loss of accuracy caused by the encoding problem. Therefore, for the improvement of the learning performance of spike-driven DSNNs, an in-depth study from the aspects of efficient deep learning construction and the spike train encoding method is required.

## 6. Conclusions

Spiking neurons encode information through precisely timed spike trains. Using the network error represented by the kernel function of spike trains, the error backpropagation algorithm BP-STIP of DSNNs is derived. In addition, in order to solve the weight transport problem existing in the error backpropagation process, the feedback alignment algorithm FA-STIP and broadcast alignment algorithm BA-STIP of DSNNs are subsequently constructed. The performance of the three algorithms is tested on the digital image MNIST dataset. The experiment results demonstrate that the two deep learning algorithms based on the feedback alignment mechanism possess better learning ability and stability. Moreover, the influence of different kernel functions and network scales on the learning accuracy of algorithms constructed by different feedback mechanisms is analyzed.

The advantages of the algorithms proposed in this paper are as follows: (1) the proposed deep learning algorithms with temporal encoding achieve a comparable classification accuracy on the MNIST dataset, and can be applied to neurons firing multiple spikes in all layers, which have biological plausibility. (2) The synaptic weight adjustment rules only depend on the spike train and its inner product operation, and do not depend on the specific dynamic of neuron models, which can be applied to DSNN with any spiking neuron models. (3) The BA-STIP learning algorithm adopts the random feedback matrices to directly propagate error signal, and can effectively reduce the consumption of computing resources during DSNN training, which is a local three-factor learning method. The main limitation of the proposed algorithms is that the learning accuracy is relatively low compared with the frequency encoding based algorithms. One principal reason is the lack of effective encoding algorithms to transform images into reasonable spike trains at this stage, which is in need of urgent in-depth research in the future.

## Figures and Tables

**Figure 1 brainsci-13-00168-f001:**
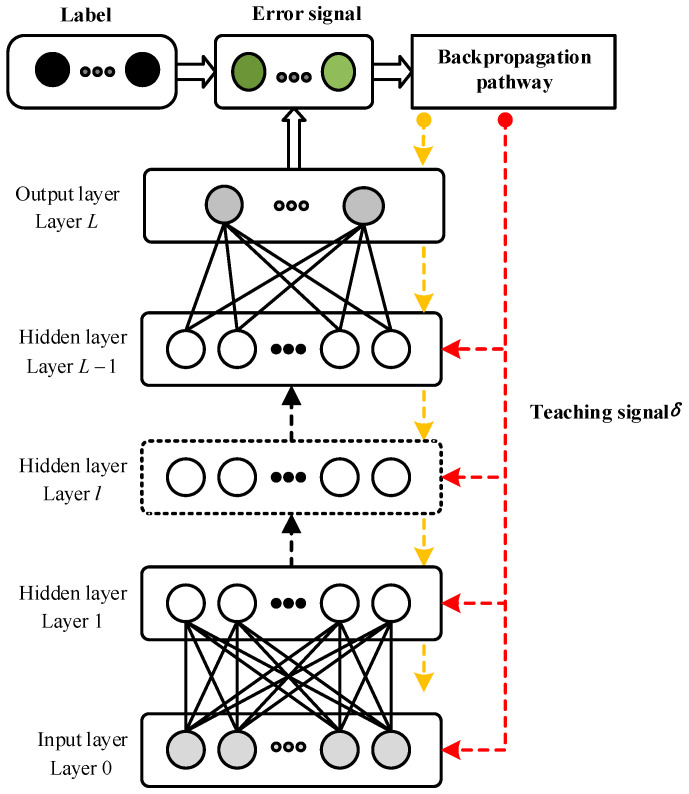
The DSNN architecture and error backpropagation pathway.

**Figure 2 brainsci-13-00168-f002:**
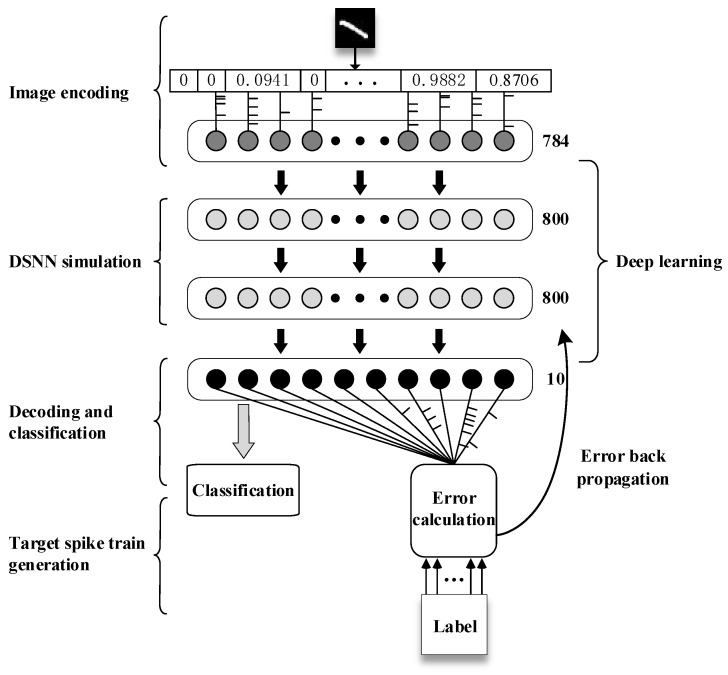
The MNIST classification computing framework based on DSNNs.

**Figure 3 brainsci-13-00168-f003:**
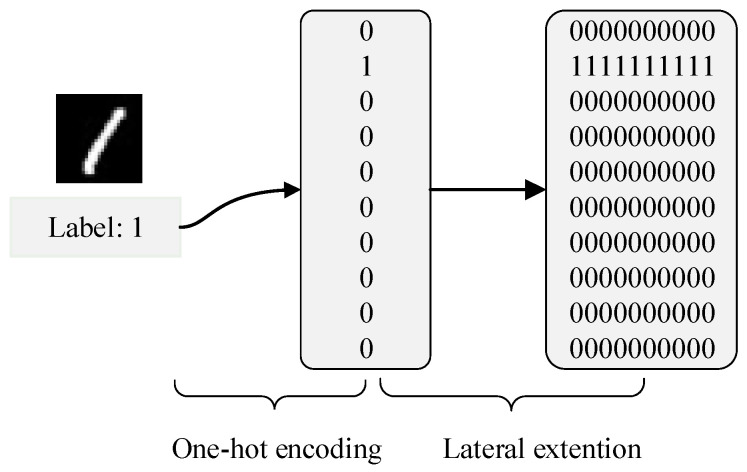
Target spike train generation process using image labels.

**Figure 4 brainsci-13-00168-f004:**
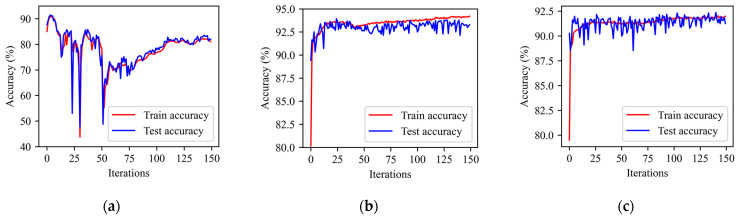
Change curve of classification accuracy of three deep learning algorithms. (**a**) BP-STIP learning algorithm; (**b**) FA-STIP learning algorithm; (**c**) BA-STIP learning algorithm.

**Figure 5 brainsci-13-00168-f005:**
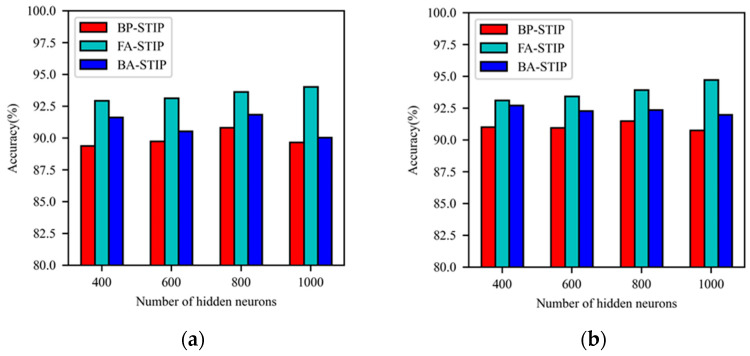
Comparison of learning performance for three STIP-based algorithms with Gaussian kernel function. (**a**) The accuracy of the training set; (**b**) the accuracy of the testing set.

**Figure 6 brainsci-13-00168-f006:**
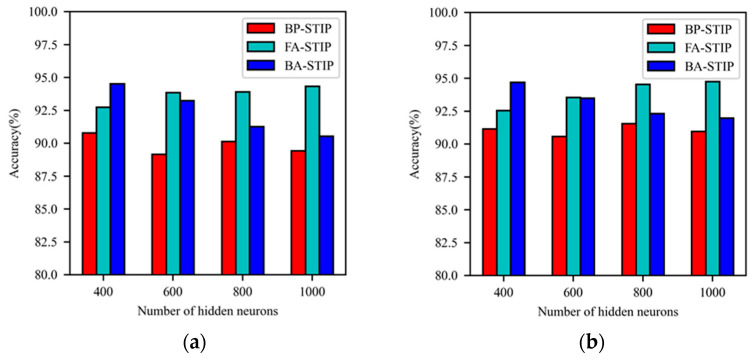
Comparison of learning performance for three STIP-based algorithms with Laplacian kernel function. (**a**) The accuracy of the training set; (**b**) the accuracy of the testing set.

**Figure 7 brainsci-13-00168-f007:**
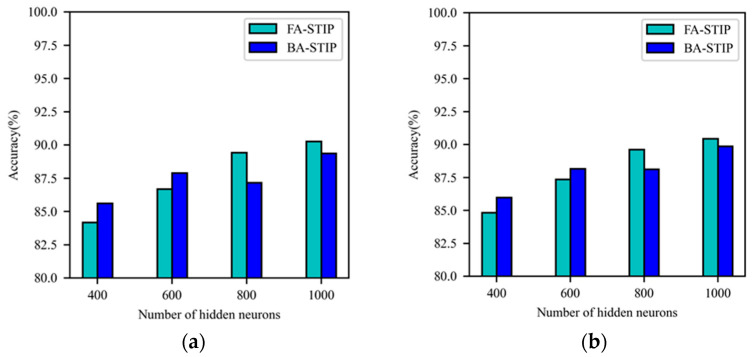
Comparison of learning performance for three STIP-based algorithms with inverse multiquadratic kernel function. (**a**) The accuracy of the training set; (**b**) the accuracy of the testing set.

**Figure 8 brainsci-13-00168-f008:**
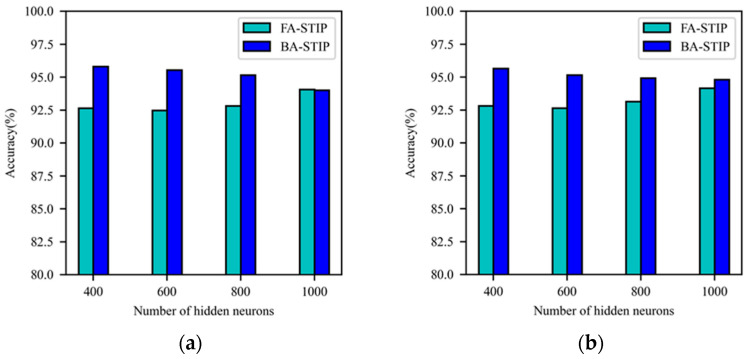
Comparison of learning performance for three STIP-based algorithms with α-kernel function. (**a**) The accuracy of the training set; (**b**) the accuracy of the testing set.

**Table 1 brainsci-13-00168-t001:** Parameter settings.

Type	Parameter	Value
Network	Number of input neurons	784
	Number of output neurons	10
	Number of hidden layers	2
	Weight initialization distribution	*N*(0,1)
LIF neuron	Time constant of membrane potential	5.0 ms
	Threshold voltage of spike firing	5.0
	Reset voltage of membrane potential	0.0
Algorithm	Random feedback path distribution	*N*(0,1)
	Batch size	64
	Learning rate	0.00008
	Epochs	150

**Table 2 brainsci-13-00168-t002:** Different kernel functions for STIP calculation.

Kernel Function	Expression	Parameter
Gaussian	$k\left(x,y\right)=\exp\left(-\frac{{\left|x-y\right|}^{2}}{2\sigma^{2}}\right)$	40
Laplacian	$k\left(x,y\right)=\exp\left(-\frac{\left|x-y\right|}{\sigma }\right)$	80
Inverse multiquadratic	$k\left(x,y\right)=\frac{1}{\sqrt{{\left|x-y\right|}^{2}+c^{2}}}$	30
α-function	$k\left(x,y\right)=\frac{\left|x-y\right|}{\sigma }\exp\left(-\frac{\left|x-y\right|}{\sigma }\right)$	5

**Table 3 brainsci-13-00168-t003:** Comparison of classification accuracy of different deep learning algorithms on MNIST dataset.

Algorithm	DSNN Structure	Neuron Model	Encoding Method	Accuracy (%)
ANN-to-SNN [23]	784-1200-1200-10	IF	frequency	98.68
ANN-to-SNN [25]	784-32C-32C-P-64C-64C-P-512-10	IF	frequency	**99.44**
ANN-to-SNN [26]	784-12C-P-64C-P-10	IF	frequency	99.09
SDBN [27]	784-500-500-10	LIF	temporal	94.09
Layer-wise STDP [29]	784-16C-16C-P-10	LIF	frequency	91.10
STDP-based Pretraining + BP [30]	784-20C-P-50C-P-200-10	LIF	frequency	99.28
BP-STDP [32]	784-500-150-10	IF	frequency	97.20
SNN-BP [35]	784-500-500-10	LIF	frequency	98.70
SNN-BP [36]	784-400-400-10	LIF	temporal	96.92
eRBP [38]	784-500-500-10	LIF	frequency	97.64
SNN-BA [39]	784-630-370-10	LIF	frequency	97.05
DeepTempo [40]	784-400-400-10	LIF	temporal	95.10
BP-STIP (Laplacian kernel)	784-800-800-10	LIF	temporal	91.56
FA-STIP (Gaussian kernel)	784-1000-1000-10	LIF	temporal	94.73
BA-STIP (α-kernel)	784-400-400-10	LIF	temporal	95.65

The highest classification accuracy of the algorithms compared on the MNIST dataset is marked in bold.

## Data Availability

The data that support the findings of this paper are openly available at the MNIST database (http://yann.lecun.com/exdb/mnist/ (accessed on 20 October 2022)).

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
