# Peer review of "Supervised Learning Algorithm Based on Spike Train Inner Product for Deep Spiking Neural Networks"

_brainsci, 2023, doi:10.3390/brainsci13020168_

Round 1

Reviewer 1 Report

This paper proposes a supervised learning algorithm based on spike train inner product for deep spiking neural networks. My comments are as follows:

1- The main contributions of this paper should be further summarized and clearly demonstrated.

2- The limitations of this work should be discussed in the experimental result and discussion section.

3 - Some new references should be added to improve the literature review—for example, https://doi.org/10.3390/app12115500; https://doi.org/10.1007/s00247-022-05510-8

Author Response

Thank you for your time in reviewing our manuscript and providing us with valuable feedback. In this response, we will aim to address your specific comments below.

Comment 0: This paper proposes a supervised learning algorithm based on spike train inner product for deep spiking neural networks. My comments are as follows.

Response: Thank you very much for your evaluation.

Comment 1: The main contributions of this paper should be further summarized and clearly demonstrated.

Response: In this paper, three supervised learning algorithms based on spike train inner products with different error feedback mechanisms for Deep Spiking Neural Networks (DSNNs) are proposed. Specifically, using the definition of kernel function and the spike trains inner product (STIP), as well as the idea of gradient-descent error backpropagation (BP), the benchmark algorithm BP-STIP is firstly proposed. In the BP-STIP algorithm, the spike train inner product is derived to represent the network error function and the synaptic weight learning rules for the different layers of DSNNs. In addition, to alleviate the weight transport problem inherent in the BP mechanism, the feedback alignment algorithm (FA-STIP) and broadcast alignment algorithm (BA-STIP) of DSNNs are further constructed. In the experiments, the effectiveness of the three learning algorithms is verified on the MNIST dataset. The experimental results show that the proposed two learning algorithms based on feedback alignment have good performance in classification accuracy and stability. We have demonstrated the main contributions of this paper more clearly in Abstract (Lines 12-24), Introduction (Lines 73-87) and Conclusions (Lines 667-690).

Comment 2: The limitations of this work should be discussed in the experimental result and discussion section.

Response: The main limitation of the proposed learning algorithms is that the classification accuracy on the MNIST dataset is lower than frequency-encoded DSNNs algorithms. Due to the lack of effective encoding algorithms to transform images into reasonable spike trains at this stage. We have discussed that in Section 5.4 (Lines 655-663) and Conclusions (Lines 688-690).

Comment 3: Some new references should be added to improve the literature review—for example, https://doi.org/10.3390/app12115500; https://doi.org/10.1007/s00247-022-05510-8.

Response: We have reviewed some recently proposed methods in Introduction and added them in the revised manuscript and numbered as [6] and [7].

Reviewer 2 Report

The authors should clarify your main contribution and why they use the image dataset. The work is a comparison with the train generation process using image labels. This can of work is well known in the image analysis field. The paper should be improved and focused on the central contribution instead of explaining the process of CNN.

Author Response

Thank you for your time in reviewing our manuscript and providing us with valuable feedback. In this response, we will aim to address your specific comments below.

Comments: The authors should clarify your main contribution and why they use the image dataset. The work is a comparison with the train generation process using image labels. This can of work is well known in the image analysis field. The paper should be improved and focused on the central contribution instead of explaining the process of CNN.

Response: In this paper, we propose three DSNN training algorithms based on different error feedback mechanisms with the help of the kernel method. In the experiments, the effectiveness of the three learning algorithms is verified on the MNIST dataset. We have demonstrated the main contributions of this paper more clearly in Abstract (Lines 12-24), Introduction (Lines 73-87) and Conclusions (Lines 667-690).

The MNIST image dataset is a widely used benchmark dataset for testing the performance of the newly proposed deep learning algorithm. As in many studies, in this paper, we also use the MNIST dataset to test the performance of the proposed learning algorithms and compare them with other related algorithms. We have explained that in Lines 401-407.

In DSNNs, information is transmitted through biologically plausible discrete spikes, which is completely different from the way in CNNs where information is transmitted through high-precision continuous floating-point numbers. In the process of DSNN recognition of MNIST images, the input images and their labels are firstly encoded into spike trains. After that, pattern classification and network weight adjustment can be realized according to the newly proposed algorithms. To clearly explain the processing steps of MNIST digital image analysis system, we divide the whole system into five submodules and described them in detail in Section 5.1 (Lines 408-441).

Reviewer 3 Report

Dear authors,

You have proposed an interesting approach based in spike neural networks. I have the following recommendations. 

1. Please add a more specific diagram of the architecture of the network, be specific in the number of layers and neurons per layer. 

2. Add in a single table all the paramters selected to train and test your model. 

3. Highlight the best results in the results table. 

4. Provide specific details about hardware and software implementation of your proposal. 

Hope these recommendations will be useful to you. 

Author Response

Thank you for your time in reviewing our manuscript and providing us with valuable feedback. In this response, we will aim to address your specific comments below.

Comment 0: Dear authors, you have proposed an interesting approach based in spike neural networks. I have the following recommendations.

Response: Thank you very much for your evaluation.

Comment 1: Please add a more specific diagram of the architecture of the network, be specific in the number of layers and neurons per layer.

Response: The benchmark network architecture for MNIST recognition is 784-800-800-10. That is to say, the network includes an input layer with 784 neurons, two hidden layers with 800 neurons, and an output layer with 10 output neurons. We have further specified the number of layers and neurons per layer in Figure 2 and explained the network architecture in Lines 464-466.

Comment 2: Add in a single table all the parameters selected to train and test your model.

Response: As you suggested, we have listed all the parameters selected to train and test our model in Table 1 in Section 5.1.

Comment 3: Highlight the best results in the results table.

Response: In Table 3, the method ANN-to-SNN [23] achieve the highest classification accuracy 99.44% on the MNIST dataset. We have marked it in bold.

Comment 4: Provide specific details about hardware and software implementation of your proposal. Hope these recommendations will be useful to you.

Response: The proposed algorithms are implemented in a Python environment using the open-source deep learning SNN framework spikingjelly based on PyTorch. All experiments were carried out on an Intel(R) Xeon(R) Gold 5218 CPU and a NVIDIA Quadro RTX 6000 GPU. We have added the hardware and software implementation of the proposed algorithms in Section 5.1 (Lines 458-461).

Round 2

Reviewer 2 Report

The results do not support the conclusions. They should be more tradable and inedible. The authors must indicate numbers. This article would benefit from close editing. Between points 3 and 3.1 should be an introductory text.

It is hard to read the article with red parts and to underline it simultaneously. Send a clean version of the paper and explain it clearly.

What is the hypothesis?

How was it checked?

what the algorithm found?

Why is it different? and why is it not a domain application?

Besides, I found it difficult to follow the author’s argument due to the many stylistic and grammatical errors.
